# Dynamic Vaccine Allocation for Control of Human-Transmissible Disease

**DOI:** 10.3390/vaccines12091034

**Published:** 2024-09-09

**Authors:** Mingdong Lyu, Chang Chang, Kuofu Liu, Randolph Hall

**Affiliations:** 1National Renewable Energy Laboratory, Mobility, Behavior, and Advanced Powertrains Department, Denver, CO 80401, USA; mingdong.lyu@nrel.gov; 2Thomas Lord Department of Computer Science, University of Southern California, Los Angeles, CA 90089, USA; cicichan@usc.edu; 3Epstein Department of Industrial and Systems Engineering, University of Southern California, Los Angeles, CA 90089, USA; kuofuliu@usc.edu

**Keywords:** COVID-19, transmission modeling, vaccine allocation

## Abstract

During pandemics, such as COVID-19, supplies of vaccines can be insufficient for meeting all needs, particularly when vaccines first become available. Our study develops a dynamic methodology for vaccine allocation, segmented by region, age, and timeframe, using a time-sensitive, age-structured compartmental model. Based on the objective of minimizing a weighted sum of deaths and cases, we used the Sequential Least Squares Quadratic Programming method to search for a locally optimal COVID-19 vaccine allocation for the United States, for the period from 16 December 2020 to 30 June 2021, where regions corresponded to the 50 states in the United States (U.S.). We also compared our solution to actual allocations of vaccines. From our model, we estimate that approximately 1.8 million cases and 9 thousand deaths could have been averted in the U.S. with an improved allocation. When case reduction is prioritized over death reduction, we found that young people (17 and younger) should receive priority over old people due to their potential to expose others. However, if death reduction is prioritized over case reduction, we found that more vaccines should be allocated to older people, due to their propensity for severe disease. While we have applied our methodology to COVID-19, our approach generalizes to other human-transmissible diseases, with potential application to future epidemics.

## 1. Introduction

Vaccination is one of the most important public health interventions for controlling the spread of infectious diseases. Vaccines work by inducing immunity to a pathogen, thereby reducing the likelihood of transmission and disease. When a portion of a population is vaccinated and acquire immunity, the likelihood of transmission decreases, as the pathogen has fewer hosts in which to replicate and spread. When a sufficient number of people acquire immunity, it is possible to reach herd immunity, which leads to diminishing rates of new infections in the population. The degree to which a population must be vaccinated to reach herd immunity depends on the transmissibility of the disease and the vaccine efficacy. In general, a higher proportion of the population needs to be vaccinated for diseases with higher transmissibility.

COVID-19 and its vaccines have led to a renewed interest in transmission modeling in epidemiology. Modeling the impact of vaccination on disease transmission is important for understanding the effectiveness of vaccination programs and designing effective vaccine distribution policies. While the immediate objective of vaccination is individual immunity against specific infectious agents, it is imperative to orchestrate campaigns to maximize benefits, including reduced risk of transmission and reduced risk of severe disease. The age of the target population is pivotal, as vaccines exhibit varied efficacy and side effects across age brackets. Rates of transmission and likelihood of severe disease also very by age. For instance, the initial Pfizer-BioNTech and Moderna vaccines demonstrated significant effectiveness across all age groups, whereas the Johnson & Johnson vaccine, while effective against severe disease and death, has shown marginally lower infection prevention rates [1].

In addition to these age-related factors, the decision-making behavior of individuals regarding vaccination can significantly influence the overall effectiveness of vaccination programs. Extensive research has explored how individual decisions can be modeled using game theory, imitation dynamics, and social network analysis. For example, Bauch and Earn (2004) and Schimit and Monteiro (2011) used game theory to demonstrate how personal and public incentives can lead to suboptimal vaccination coverage if individuals act based on self-interest [2,3]. Bauch (2005) and Fu et al. (2011) further highlighted how social influence and imitation dynamics within social networks drive vaccination behavior, where individuals are likely to follow the actions of their peers [4,5]. Klepac et al. (2016) and Reluga et al. (2006) emphasized the importance of collective action and public perception, showing that cooperation among regions and positive public opinion are crucial for maintaining high vaccination rates [6,7]. Arefin et al. explored how the structure of social networks can lead to inefficiencies in vaccination outcomes, pointing to the potential for network-based interventions [8]. These studies collectively underscore the complex interplay between individual behavior, social dynamics, and public health policies in determining vaccination coverage.

In our study, we focused on the allocation of limited vaccine supplies during a period where distribution, rather than demand, was the primary challenge. Therefore, individual decision making was not incorporated into our model. However, we recognize its importance in broader vaccination strategies. Our paper focuses on the allocation of vaccines both across regions and age group at times when they are in short supply. When vaccines initially became available in the U.S. in 2021, individuals at the highest risk of severe outcomes, encompassing the elderly and individuals with comorbidities, were prioritized, as were healthcare professionals, due to risk of exposure as well as criticality of occupations. On the other hand, all regions of the U.S. received allocations of vaccines without explicitly considering whether the regions were experiencing high or low rates of infections.

In this paper, we introduce a transmission model accounting for age and vaccination status within regions, leveraging dynamic modeling with time-varying parameters to depict disease transmission dynamics. Validated against historical data from all 50 states, our model captures disease transmission nuances, offering insights for health policy decisions. We further propose an innovative vaccine allocation method, accounting for regional disparities in transmission, population structure, and resource limitations. This dynamic allocation method seeks to efficiently mitigate disease impact. By analyzing various vaccine allocation policies under diverse availability scenarios, our age-structured dynamic model offers insights into effective disease management strategies, guiding vaccine distribution. Our research aims to deepen the understanding of disease transmission with mathematical modeling while promoting equitable and efficient resource utilization.

## 2. Previous Research

This section assesses the role of vaccination in infectious disease models, using various case studies for illustration, with a focus on COVID-19 in particular. We also discuss the challenges and strategies in vaccine allocation during epidemics. Notably, while current models provide significant insights, there remains a gap in addressing the dynamic nature of disease transmission and demographic variations. This introduction serves as a roadmap for the subsequent detailed exploration in Section 3.

### 2.1. Dynamic Transmission Models with Vaccination Consideration

The strategic use of vaccinations is pivotal in mitigating pandemics. For accuracy, disease transmission models must dynamically represent vaccinated individuals over time and account for real-world vaccine characteristics. This allows the models to depict both the direct protection conferred upon vaccinated individuals and the indirect effects on the broader population. Much of the available research draws on compartmental modeling, through which a population is divided into groups represented by their common characteristics. Compartmental modeling utilizes differential equations to represent transitions of individuals between compartments, such as when people become infected or when they recover from a disease [1]. Several studies have pioneered ways to represent vaccinated groups and amalgamate vaccine characteristics into transmission models. Alexander et al., for instance, compartmentalized the population into susceptible (S), infected (I), recovered (R), and vaccinated (V) groups [9]. Some models even introduce an “exposed” component [10]. This compartmental approach enables tracking of both vaccinated and unvaccinated individuals, taking into account the fact that vaccines might not offer complete protection. Bai, Song, and Xu’s model from 2021 incorporated the concept of waning vaccine-induced immunity, which sees vaccinated individuals eventually reverting to susceptibility [11]. More sophisticated models introduce delays between latent and symptomatic stages of disease, combined with waning immunity. Such structures represent real-world observations where vaccinated individuals might contract the disease but show slower progression [12]. Wu’s model, analyzing the pneumococcal vaccine’s cost effectiveness in Taiwan, distinguished between infected vaccinated and unvaccinated individuals, attributing different hospitalization and mortality rates to each group [13]. The model also featured a vaccine match parameter, offering insights into scenarios of well-matched versus poorly matched vaccines. Considering the diverse nature of human populations, many models factor in heterogeneity and transmission dynamics. Ko et al. acknowledged this diversity in their study, classifying the population based on age, occupation, and health status, among other factors [14]. Monod’s framework married human mobility and mortality rates, using age-specific contact patterns to dissect disease transmission dynamics [15]. Other research has stratified the population into risk groups and then further segment them into susceptible, vaccinated, infected, and recovered categories [16]. The underlying assumption in some studies is that vaccination directly transitions an individual from susceptibility to full recovery, presuming a perfect vaccine [17]. Other models emphasize the dynamics between vaccine efficacy, coverage, and disease transmission. Bubar et al. used a mathematical model to compare age-stratified prioritization strategies for the SARS-CoV-2 vaccine. They found that prioritizing vaccines for adults aged 20–49 minimized cumulative incidence, while prioritizing adults over 60 minimized mortality and years of life lost in most scenarios. They also considered individual-level serological tests to improve the impact of each dose, reflecting real-world conditions and potential disparities in COVID-19 impact. However, while the inclusion of serological testing improves the precision of vaccine allocation, it also introduces complexities related to the availability, accuracy, and timeliness of serological tests [18].

### 2.2. Vaccine Allocation Methods

The foundation of effective vaccine allocation lies in robust disease transmission models. These models utilize optimization techniques that account for real-world challenges, such as limited vaccine availability, dosing guidelines, daily vaccination capacities, and efficacy. Due to the complexities of the underlying disease model, it is at present impossible to guarantee global optimality for an objective of minimizing total number of infections or total number of fatalities. In place of global optimality, heuristics and localized search methods have been studied in a variety of publications.

Hill and Longini established a mathematical framework to determine the minimal vaccine doses necessary to curtail an outbreak [19]. Central to their work is the concept of “herd immunity”, aiming to maintain an effective reproductive number below one. Key factors in establishing this threshold are the initially susceptible population and the basic reproductive number [20,21,22]. Techniques like the “Minimum Dose with Satisfactory Efficacy” (MDSE) approach aim to achieve optimal protection levels using minimal vaccine doses, grounded in empirical data [23].

Recognizing the inherent constraints in vaccine production, research has evolved to focus on optimizing public health benefits with limited resources. Medlock and Galvani examined the optimization of influenza vaccine distribution. They utilized a mathematical model to determine the most effective distribution of vaccines to minimize the overall influenza cases. Their findings suggest that prioritizing schoolchildren and adults aged 30 to 39 years could be more effective than the traditional approach of prioritizing the elderly and high-risk populations [24]. In the context of the COVID-19 pandemic, Meehan and colleagues used an age-structured mathematical model to investigate the benefits of optimizing age-specific dose allocation for the COVID-19 vaccine. Their results indicate that prioritizing individuals between 30 and 59 years of age can minimize transmission due to their high contact rates and increased risk of infection. However, to effectively reduce morbidity and mortality, targeting those aged 60 and above is crucial as they are more susceptible to severe disease [25]. Additionally, Gonz’alez-Parra and team presented two nonlinear mathematical models to understand the optimal vaccination strategy considering the case fatality rate and age structure of a population. Their findings suggest that in most scenarios, the best approach is to prioritize vaccination for individuals aged 55 and above. However, under certain conditions, such as high transmission rates, it might be more effective to first allocate vaccines to the 15–54 age group [26].

In the paper “Where to locate COVID-19 mass vaccination facilities?” by Bertsimas et al. [27], the authors address the complex challenge of optimally distributing the COVID-19 vaccine, in application to the U.S. Utilizing a data-driven approach, they enhance the epidemiological model DELPHI to factor in vaccination effects and mortality rate variability across age groups. This augmented model then underpins a prescriptive model aimed at identifying optimal vaccination site locations and vaccine allocation strategies. The problem is formulated as a bilinear, nonconvex optimization model, which they tackle using a coordinate descent algorithm. Their methodology, when benchmarked, potentially improves vaccination campaign effectiveness by an estimated 20%, equating to around 4000 additional lives saved in the U.S. over three months. However, it is essential to recognize that the practical effectiveness of such models often depends on the availability of real-time data, which can be missing or delayed in real-world situations, potentially impacting their applicability [9].

One important study segmented a population based on age-related risks, evaluating various vaccination strategies against metrics like infection rates and healthcare preparedness [26,28]. Valizadeh’s research introduces a robust bi-level optimization model to tackle challenges in the COVID-19 vaccine supply chain. When tested in Kermanshah, Iran, the model effectively minimized mortality risk, distribution inequality, and overall costs, providing insights for improved vaccination management [29].

### 2.3. Summary and Research Gap

Historically, epidemiological research has shown the importance of vaccination in controlling infectious diseases, ranging from hepatitis B to COVID-19. Mathematical models simulating transmission dynamics, with varying granularity in vaccination parameters, have facilitated strategic decision making and efficient resource allocation. Nonetheless, existing models could benefit from the incorporation of time-dependent transmission parameters and the nuanced effects of age demographics on disease transmission. For instance, Ko’s age-stratified model for COVID-19 underscores the need for a more differentiated approach to vaccine allocation, considering the unique dynamics of different age groups [14]. Though the domain of vaccine allocation optimization has seen substantial progress, the prevailing models could be augmented by considering the dynamic nature of disease spread. A spatially sensitive approach, accounting for differential transmission rates among regions, would bolster targeted vaccine allocation, thereby maximizing the societal benefit.

Building on the existing literature, our study bridges several gaps. First, our method for vaccine allocation, described in Section 3, considers transmission rates and age demographics by region along with constrained numbers of vaccine doses.

Second, our transmission model uses a continuous Sigmoid function to represent dynamic changes in death and transmission rates, reflecting the hypothesis that these rates change continuously due to evolving changes in human behavior. Third, we are able to solve the complex model to local optimality for the U.S., allocating vaccines to the 50 states and three age groups, over seven-month periods. Utilizing reported data on cases, deaths, and vaccine availability, we demonstrate how the model can be applied from the types of data which were routines collected in the U.S. during the COVID-19 pandemic.

Our comprehensive scenario analysis underscores the potential for adaptable vaccine distribution strategies, especially in resource-limited contexts. We target allocation both demographically and spatially, aiming to guide efficient and equitable resource utilization in the face of infectious diseases.

## 3. Methodology

This section develops our disease model and explains our solution process. The model includes multiple components, which are developed from reported data for COVID-19. Our model represents the interplay between age-specific transmission dynamics, vaccine rollout, and disease outcomes. By considering the heterogeneous nature of COVID-19 transmission and death rates across different age groups and accounting for the impact of vaccination, we can generate more accurate and nuanced predictions about the pandemic’s progression. This, in turn, can help inform public health policies and interventions tailored to the specific needs of each age group, ultimately contributing to more effective disease control. Initially, an age-structured SEIRD model is introduced, compartmentalizing individuals into the states: susceptible, exposed, infected, recovered, and dead. These transitions are described using differential equations, highlighting the model’s adaptability in capturing time-sensitive nuances in transmission and death rates. The population is then segmented into distinct age groups, each exhibiting unique transmission and mortality characteristics. The granularity is further enhanced by introducing vaccination data into the model, revealing the relationship between age-based transmission, vaccine rollout, and disease outcomes. To capture the evolving nature of transmission dynamics, a Sigmoid function is employed, offering a versatile tool to track changes in transmission and death rates over time. Moving from modeling to decisions, we develop a dynamic framework for vaccine allocation across the U.S. states. Our framework utilizes age-structured data and accounts for the effects of vaccination. The model’s nonlinearity, due to implicit ordinary differential equations, is captured with the Sequential Least Squares Quadratic Programming method, which is used to search for a locally optimal vaccine allocation.

### 3.1. Data Sources

In this study, we leveraged data from the COVID Tracking Project [30] and Our World in Data [31], which provide daily and cumulative figures for confirmed cases, deaths, and demographic information related to COVID-19. We sourced vaccination data from the CDC’s COVID-19 Vaccine Distribution Allocations by Jurisdiction [32], which includes total distribution and administration figures for the Janssen, Moderna, and Pfizer vaccines, segmented by jurisdiction and age group. Our analysis spanned from 16 December 2020 to 30 June 2021. It should be noted that, prior to 12 February 2021, the CDC’s records only reflected the total quantity of vaccines administered state-wide, without age group differentiation. To address this data gap, we approximated the distribution of vaccines among different age groups by applying the proportional distribution of eligible individuals within each age category, as stipulated by the prevailing vaccination guidelines at the time.

### 3.2. Age-Structured Dynamic Model with Vaccination

Based on the SEIRD compartmental model, we categorized the population into five groups: susceptible (S), exposed (E), infected (I), recovered (R), and dead (D) [33,34]. This model, preferred for its adaptability and simplicity, tracks transitions between these states over time, using differential equations. Initially, everyone is considered susceptible. Upon contact with an infected person, susceptible people can become exposed and eventually infected after the incubation period. People then either recover and gain immunity or succumb to the disease. Chosen for its inherent flexibility, the SEIRD model offers insights without the necessity for detailed individual data which are often lacking during an epidemic’s onset.

The transmission of the disease is governed by the transmission rate β(t), which represents the rate of an infected person transmitting the disease to a susceptible person. The transmission rate is typically assumed to be constant in the basic SEIR model, but it is modeled as a time-varying parameter to capture changes in behavior or the impact of interventions on disease transmission. Death rate α(t) is also treated as a time-varying function, representing the proportion of infectious people who eventually die from the disease, by date. Those who eventually die transition from an infected to a deceased state at a rate of *ρ*, representing the inverse of the time from becoming infectious until the time of death. In our model, *ρ* is assumed to be constant over time. Those who eventually recover do so at the rate γ, representing the inverse of the time from becoming infectious until recovery. We will also later derive the effective reproduction number Rep(t), representing the average number of persons who are exposed to the disease by each infectious person, as a function of time.

To analyze vaccination’s influence on COVID-19 dynamics, we segment the population into three age groups: 0–17, 18–64, and 65+. Each group exhibits distinct transmission and mortality rates. We introduce
$\omega_{i}$ and
$\tau_{i}$ for each age group, quantifying relative transmission and mortality risks compared to reference groups.

Vaccination is incorporated via a vaccine compartment Vi for each age group, adjusting the number of susceptible individuals based on vaccination rate and vaccine effectiveness θ. This accounts for potential infections among vaccinated individuals. However, our model omits the vaccine’s impact on death rates due to data limitations and excludes interstate movement because of unavailable age-specific mobility data. The resulting SEIRD-V model is detailed in Equation (1).
(1)$$\begin{array}{cc} & \frac{\partial S_{i}\left(t\right)}{\partial t}=-\omega_{i}\cdot \beta \left(t\right)\cdot I_{i}\left(t\right)\cdot \frac{S_{i}\left(t\right)-\theta \cdot V_{i}\left(t\right)}{N}\\ & \frac{\partial E_{i}\left(t\right)}{\partial t}=\omega_{i}\cdot \beta \left(t\right)\cdot I_{i}\left(t\right)\cdot \frac{S_{i}\left(t\right)-\theta \cdot V_{i}\left(t\right)}{N}-\sigma \cdot E_{i}\left(t\right)\\ & \frac{\partial I_{i}\left(t\right)}{\partial t}=\sigma \cdot E_{i}\left(t\right)-\left(1-\tau_{i}\cdot \alpha \left(t\right)\right)\cdot \gamma I_{i}\left(t\right)-\tau_{i}\cdot \alpha \left(t\right)\cdot \rho \cdot I_{i}\left(t\right)\\ & \frac{\partial R_{i}\left(t\right)}{\partial t}=\left(1-\tau_{i}\cdot \alpha \left(t\right)\right)\cdot \gamma \cdot I_{i}\left(t\right)\\ & \frac{\partial D_{i}\left(t\right)}{\partial t}=\tau_{i}\cdot \alpha \left(t\right)\cdot \rho \cdot I_{i}(t) \end{array}$$ where


$S_{i}\left(t\right)$: number of susceptible individuals in age group
i over time;$E_{i}\left(t\right)$: number of exposed individuals in age group
i over time;$I_{i}\left(t\right)$: number of infectious individuals in age group
i over time;$R_{i}\left(t\right)$: number of recovered individuals in age group
i over time;$D_{i}\left(t\right)$: number of dead individuals in age group
i over time;$V_{i}\left(t\right)$: number of individuals vaccinated in age group
i over time;N: total population size;β(t): effective contact rate, a measure of how many people to whom an infected person can transmit the disease at time t;α(t): fraction of infectious individuals detected and isolated at time t;γ: recovery rate of infected individuals;δ: rate at which exposed individuals become infectious;*ρ*: fatality rate among infected individuals;$\omega_{i}$: scalars representing the difference in transmission rate between age groups
i with respect to age group 2;$\tau_{i}$: scalars representing the difference in fatality rate between age groups
i with respect to age group 3.


$\alpha \left(t\right)$ and
$\beta \left(t\right)$ can vary over time due to changes in human behavior. A natural function for representing patterns of change is the Sigmoid function. Equation (2) is the general form of the Sigmoid function, where
k determines the slope of the function and
a determines the *x* value at the middle point (i.e., the point of time when *y* = 0.5) [35].
(2)$$S\left(x\right)=\frac{1}{1+e^{k\left(x-a\right)}}$$

Thus, we define the function for transmission rate and death rate as Equations (3) and (4):
(3)$$\beta \left(t\right)=\beta_{end}+\frac{\beta_{start}-\beta_{end}}{1+e^{m\cdot \left(x-a\right)}}$$
(4)$$\alpha \left(t\right)=\alpha_{end}+\frac{\alpha_{start}-\alpha_{end}}{1+e^{n\cdot \left(t-b\right)}}$$ where


$\beta_{start}$ is the starting reproduction number;$\beta_{end}$ is the ending reproduction number;$\alpha_{start}$ is the starting death rate, ranging from 0 to 1;$\alpha_{end}$ is the ending death rate, ranging from 0 to 1;m, n, a, and
b are shape parameters.


Parameters will be estimated with the objective of minimizing the weighted summation of squared error between cumulative predicted and measured confirmed cases and the summation of squared error between cumulative predicted and cumulative confirmed deaths. The best fitted value for each parameter is solved by the Levenberg–Marquardt algorithm (LMA), which is a combination of the steepest descent method and the Gauss–Newton method [36]. In Section 4, we will present the model’s accuracy for the period from 16 December 2020, the date of the first vaccine administration, until 30 June 2021, covering all 50 American states.

### 3.3. Vaccine Allocation with Dynamic Transmission

In this section, we propose a dynamic framework to allocate vaccines among 50 states in the United States, taking into account the transmission patterns and the impact of vaccination on disease transmission and death rates.

#### 3.3.1. Problem Formulation

In Section 3.2, we developed a transmission model incorporating vaccination data to provide reasonably accurate estimates of COVID-19 cases and deaths across different age groups and states. The model considers age-structured case and death data, vaccine data, and time-varying transmission and death rates, accounting for the effects of vaccination on susceptible populations in each age group. The fitting results demonstrated that the model effectively captured the historical trends of COVID-19 cases and deaths and the impact of vaccination on these trends.

Building upon this transmission model, we now aim to allocate vaccines among the 50 states. The objective function is defined as the sum of the weighted case numbers and death numbers. The constraints are the biweekly available amount of vaccination for each state. Equation (5) provides our formulation:
(5)$$\begin{array}{cc} & \underset{V_{i,t}\;}{\min}\sum_{i,t}w_{1}\cdot Cases_{i,t}+w_{2}\cdot Deaths_{i,t}\\ & s.t.\sum_{i}V_{i,t}\leq Q_{t}\forall t\\ & \;\;\;0\leq V_{i,t}\leq N_{i}\\ & \;\;\;\frac{\partial S_{i}\left(t\right)}{\partial t}=-\omega_{i}\cdot \beta \left(t\right)\cdot I_{i}\left(t\right)\cdot \frac{S_{i}\left(t\right)-\theta \cdot V_{i}\left(t\right)}{N}\\ & \;\;\;\frac{\partial E_{i}\left(t\right)}{\partial t}=\omega_{i}\cdot \beta \left(t\right)\cdot I_{i}\left(t\right)\cdot \frac{S_{i}\left(t\right)-\theta \cdot V_{i}\left(t\right)}{N}-\sigma \cdot E_{i}\left(t\right)\\ & \;\;\;\frac{\partial I_{i}\left(t\right)}{\partial t}=\sigma \cdot E_{i}\left(t\right)-\left(1-\tau_{i}\cdot \alpha \left(t\right)\right)\cdot \gamma I_{i}\left(t\right)-\tau_{i}\cdot \alpha \left(t\right)\cdot \rho \cdot I_{i}\left(t\right)\\ & \;\;\;\frac{\partial R_{i}\left(t\right)}{\partial t}=\left(1-\tau_{i}\cdot \alpha \left(t\right)\right)\cdot \gamma \cdot I_{i}\left(t\right)\\ & \;\;\;\frac{\partial D_{i}\left(t\right)}{\partial t}=\tau_{i}\cdot \alpha \left(t\right)\cdot \rho \cdot I_{i}\left(t\right)\\ & \;\;\;Cases_{i,t}=I_{i}\left(t\right)+R_{i}\left(t\right)+D_{i}\left(t\right)\\ & \;\;\;Deaths_{i,t}=D_{i}(t) \end{array}$$ where
$V_{i,t}$ refers to the vaccines administered in region
i on day
t;
$Q_{t}$ refers to the total amount of available vaccines on day
t;
$w_{1}$ and
$w_{2}$ are the weights assigned to the number of cases and deaths in the objective function. The dynamic nature of the formulation lies in the fact that it considers the evolving transmission patterns and vaccination rates over time.

#### 3.3.2. Solution Process

We utilized the Sequential Least Squares Programming optimizer (SLSQP) method [37], a gradient-based optimization algorithm, for vaccine allocation. The SLSQP algorithm is well suited for our problem because it can handle both equality and inequality constraints and is capable of solving nonlinear optimization problems with a large number of variables.

To illustrate the SLSQP solving process, let us first consider the optimization problem, which aims to minimize the weighted sum of cases and deaths over a specific time horizon. The decision variables are the biweekly vaccine allocations for each state and age group, subject to constraints on the total available vaccines and the maximum vaccination capacity of each state. The SLSQP algorithm starts with an initial guess for the decision variables (i.e., the biweekly vaccine allocations) and iteratively updates these values to reduce the objective function. At each iteration, the algorithm computes the gradient of the objective function with respect to the decision variables, which is essential for updating the decision variables in the right direction.

In this context, the gradient computation is challenging due to the implicit nature of the objective function, which depends on the solution of ordinary differential equations (ODEs) describing the transmission dynamics. To calculate the case/death numbers in the objective functions and the gradient with respect to the decision variables, we must first solve the ODEs. Traditional third-party ODE solvers, such as the odeint function provided by the SciPy library, utilize the fourth-order Runge–Kutta method for accuracy. This method approximates the daily increments with a sufficiently small step size. However, the transmission rate and death rate of the disease will remain constant within the same day, and the implicit formulation of the fourth-order Runge–Kutta method makes it challenging for the algorithm to find the derivatives with respect to the decision variables, potentially leading to gradient vanishment.

To address this issue, we utilized Euler’s method, a first-order numerical method for solving ODEs, instead of the fourth-order Runge–Kutta method. By employing Euler’s method, we can calculate the weekly case and death numbers with fixed decision variables (i.e., biweekly allocated vaccination for each state). This approach allows for a more straightforward computation of the gradient, avoiding the complexities associated with higher-order ODE solvers like the Runge–Kutta method.

Once the gradient is computed, the SLSQP algorithm updates the decision variables by moving in the direction of the negative gradient, which corresponds to the steepest descent in the objective function. The algorithm also takes into account the constraints on vaccine availability, ensuring that the updated decision variables are feasible. This iterative process continues until the improvement in the weighted sum of cases and deaths is smaller than the threshold, indicating that further iterations are unlikely to significantly enhance the solution. A schematic of the solution process is shown in Figure 1.

The SLSQP-based optimization framework offers a systematic approach for searching for an optimized distribution of vaccines among the 50 regions and 3 age groups, taking into account the dynamic nature of transmission patterns, vaccine availability, and state capacities. The dynamic vaccine allocation algorithm produces an improved distribution of vaccines, considering the regional transmission patterns and the impact of vaccination on disease transmission and death rates.

In summary, Euler’s method helps solve the highly nonlinear optimization problem with implicit ODEs. The SLSQP method for solving the optimization problem ensures that the algorithm can find the derivatives with respect to the decision variables, enabling an effective solution method. By incorporating regional transmission patterns and the impact of vaccination on disease transmission and death rates, the proposed framework enables a more targeted and efficient allocation of vaccines [36].

## 4. Results

We first present the accuracy of the age-structured dynamic model’s predictions, focusing on COVID-19 transmission across three pivotal age groups using vaccination data. Model accuracy provides insights into state-wise and age-specific pandemic trends, reflecting variations influenced by factors like vaccination rates and public health adherence. We next provide our vaccine allocation results under varying vaccine availability scenarios. Through examination of diverse allocation methods, we analyze the specific outcomes tied to each strategy. The scenarios range from a hypothetical absence of vaccination to a potential tenfold increase in vaccine availability.

### 4.1. Fitting Results

Figure 2 and Figure 3 summarize the model fitting accuracy of the transmission model with vaccination for three age groups (0–17, 18–64, and 65+) across all 50 states in the United States from 16 December 2020 to 30 June 2021. Figure 3 shows the fitting results of case/death number across three age groups for California and New York. The model’s accuracy is measured by the relative root-mean-square error:

(RRMSE) (defined as RRMSE =
$\frac{{\left[\sum_{i=1}^{N}{\left(\hat{y_{i}}-y_{i}\right)}^{2}/N\right]}^{1/2}}{y_{N}}$ for both the number of COVID-19 cases and deaths within each age group for each state).

On average, the RRMSE for the number of cases is 0.092 for the 0–17 age group, 0.080 for the 18–64 age group, and 0.078 for the 65+ age group. The average RRMSE for the number of deaths is 0.009 for the 0–17 age group, 0.073 for the 18–64 age group, and 0.038 for the 65+ age group.

We note that the RRMSE is sometimes lower for cases than deaths. This is a byproduct of situations where the reported deaths were very few, sometimes even zero, in certain states, age groups, and time periods. Such low counts can produce high RRMSE values, even when absolute errors are low. For example, in the 0–17 age group, COVID-19-related deaths were rare throughout the country and therefore hard to predict within a small percentage error.

Figures Figure 4 shows the fitting results of case/death numbers across three age groups for California and New York. The lower fitting accuracy for certain states may also be attributed to the discrete daily variation in administered vaccines. Daily fluctuations in vaccination numbers can add complexity to the modeling process, making it more challenging for the model to generate smooth transmission rate and death rate functions that accurately capture historical trends. This issue can be particularly pronounced in states with inconsistent vaccination rollouts or disruptions due to supply chain issues, logistical challenges, or changes in vaccine eligibility criteria. In such cases, the model may struggle to accurately account for the impacts of these fluctuations on overall transmission and death rates. The daily variation in administered vaccines can lead to inconsistencies in the model’s predictions, which may contribute to lower fitting accuracy observed in some states.

In summary, the model fitting results demonstrate that incorporating vaccination data into a transmission model can provide reasonably accurate estimates of COVID-19 cases and deaths across different age groups and states. The model’s varying accuracy across states highlights the importance of considering regional factors when evaluating its performance and potential improvements. Despite its limitations, the transmission model with vaccination data offers valuable insights into the progression of the COVID-19 pandemic in the United States, particularly in terms of age-specific trends. By accounting for the effects of vaccination and different age group transmission dynamics, our model can help inform public health policies and interventions that are tailored to the specific needs of each age group.

### 4.2. Vaccine Allocation under Different Scenarios

In this section, we provide the results for vaccine allocation under varied vaccine availability scenarios, addressing the period from 16 December 2020 to 30 June 2021 in the U.S. We allocate vaccines by U.S. state, time interval, and age group; the age groups are 0–17 years, 18–64 years, and 65+ years. Although youth and children were not initially eligible for vaccination, our analysis considers the benefits that would have occurred if they were eligible at an earlier time. The study period, representing a time when vaccines were in short supply, was divided biweekly (i.e., the vaccine is distributed among states biweekly). By exploring diverse vaccine allocation methods, we aim to understand the ramifications of different prioritizations and vaccine quantities. Evaluating different distribution scenarios allows us to weigh the advantages and disadvantages of prioritizing specific age groups or regions versus focusing on broader coverage. We included analyses of these scenarios:No vaccines.Vaccine allocation with supplies matching actual availabilities, where (a) case reduction was prioritized, by setting w_2_ = 0, or (b) death reduction was prioritized, by setting w_1_ = 0.Vaccine allocation with 10 times the actual availability.

By starting with a scenario of zero vaccine availability, we establish a baseline to gauge the effectiveness of actual vaccine allocations. This baseline understanding is instrumental in discerning the merits of vaccine policies. To explore the best vaccine distribution policy, we can consider different scenarios of vaccine availability. For each scenario, the age-structured dynamic model with vaccination can be used to simulate the impact of various distribution strategies on the number of cases and deaths.

Our simulations and allocations were run on Google Colab, with the hardware of Intel Xeon CPU with two vCPUs (virtual CPUs) and 13GB of RAM. The simulation of the no-vaccination scenario took approximately 312 min to run 25 iterations. The allocation simulations for case reduction and death reduction with original and 10-time vaccine allocation policies took 370 min and 420 min to run 25 iterations.

#### 4.2.1. Healthcare Outcomes without Vaccination

The hypothetical situation of zero vaccines allows us to understand the effectiveness of the historical vaccine distribution by comparing the case and death numbers under this scenario to the real historical data. By simulating the age-structured dynamic model with no vaccination, we can estimate the number of cases and deaths that would have occurred if no vaccines were distributed.

The results from this analysis show that the historical vaccine distribution has had a significant impact on reducing the spread of viruses and saving lives. In the absence of any vaccine distribution, the model estimates that there would have been an additional 1,827,631 cases and 9180 deaths between 16 December 2020 and 30 June 2021. Our findings highlight the crucial role that vaccines have played in mitigating the severity of the pandemic and demonstrate the importance of an effective vaccine distribution strategy.

#### 4.2.2. Vaccine Allocation Policy with Original Vaccine Availability

In the second scenario, we explore the re-allocation of a constant quantity of vaccines among the 50 states and among age groups to achieve better healthcare outcomes while considering the dynamic transmission patterns in each state. To address this issue, we analyzed vaccine allocation under two different prioritizations: reducing the number of cases as much as possible and reducing the number of deaths as much as possible.

When focusing on case reduction, a larger share of vaccines was distributed to the youngest age group (0–17 years), as shown in Figure 5. This is because younger people are generally more active and have more frequent social interactions, leading to a higher potential for spreading the virus. Additionally, younger individuals may exhibit milder symptoms or be asymptomatic, making them more likely to unknowingly transmit the virus to others. By prioritizing this age group, the overall transmission rate within the population can be significantly reduced, ultimately lowering the total number of cases. In this prioritization, the focus is on reducing the spread of the virus, leading to an overall decrease in cases and, consequently, a lower number of associated deaths. According to the model results, this allocation strategy reduced 2,042,312 cases and 1796 deaths.

When prioritizing reduction in deaths, more vaccines are allocated to both the younger age group (0–17 years) and the older population (65+ years), shown in Figure 6 who face a higher risk of severe illness and death from COVID-19. The middle age group receives fewer vaccines. While this allocation strategy prioritizes death reduction, cases as are also reduced because even a reduction in cases among the young ultimately saves the lives of vulnerable people, who are less likely to contract the disease. According to the model results, this allocation strategy could reduce 220,010 cases and 6319 deaths.

The results of the second scenario demonstrate that a more targeted vaccine allocation strategy, considering the dynamic transmission patterns, can lead to significantly better health outcomes. By shifting the vaccine distribution towards age groups that have the most significant impact on transmission and death rates, it is possible to achieve substantial reductions in both cases and fatalities.

Figure 7 and Figure 8 show the redistribution of original amount of vaccine among 50 states for case-prioritized and death-prioritized allocations. Some states, like Maine, Vermont, Montana, South Dakota, and New Hampshire, receive more vaccines. These states generally have smaller populations, fewer resources, and limited healthcare infrastructure, particularly in rural areas, which can affect their ability to quickly identify, treat, and manage cases. An increased allocation of vaccines could help to compensate for these limitations by reducing the number of severe cases that require hospitalization and specialized care. Additionally, the age distribution of the population in these states may play a role in the increased need for vaccines, as some have a higher proportion of older adults who are at greater risk of severe illness and death due to COVID-19. Prioritizing vaccine allocation to these states can help protect their most vulnerable citizens and reduce fatalities. Lastly, the effectiveness of public health policies and their implementation varies between states, and those with a higher need for vaccines may have less stringent public health measures or lower compliance.

#### 4.2.3. Vaccine Allocation Policy with 10 Times the Original Vaccine Availability

In the third scenario, we explore the impact of a substantial increase in vaccine availability, specifically 10 times the original weekly availability. This scenario aims to understand the optimal vaccine allocation policy and the corresponding healthcare outcomes, given this significant increase in resources. Similar to the second scenario, we analyze the best vaccine allocation policy under two different prioritizations: reducing the number of cases as much as possible and reducing the number of deaths as much as possible.

The vaccine allocation results for both prioritizations remain consistent with the second scenario, as shown in Figure 9 and Figure 10. To reduce cases, vastly more vaccines are allocated to the younger age group (0–17 years) due to their role in driving overall case numbers. Additionally, prioritizing the vaccination of the older population is recommended if policymakers emphasize a reduction in fatalities. This demonstrates the robustness of the allocation strategies across different levels of vaccine availability.

According to the model results, the allocation strategy could potentially reduce 2,561,885 cases and 6735 deaths when prioritizing a reduction in cases. Although this represents a significant reduction in case numbers, the increment in vaccination resources only leads to an additional reduction of 519,573 cases. This is primarily because, even with 10 times the vaccine availability for each week, the available amount of vaccine is still too small to control the epidemic during the first several months. The limited vaccination resources at the beginning of a new wave of transmission makes it difficult to substantially reduce the number of cases once the disease has spread widely throughout the population.

In contrast, prioritizing deaths produces an estimated reduction of 1,537,008 cases and 16,014 deaths, leading to an additional 9695 lives saved. These finding highlights that, even though a 10-fold increase in vaccines may not result in a dramatic decrease in case numbers, it can still have a substantial impact on saving lives. The allocation of sufficient vaccines can protect vulnerable populations, particularly the older population, who are at the highest risk of severe illness and death from COVID-19.

When analyzing the third scenario, it is crucial to acknowledge the inherent limitations of increasing vaccine availability. A 10-fold increase in weekly availability may not be feasible due to production constraints, logistical challenges, and the need for a rapid and efficient rollout. Nonetheless, this scenario provides valuable insights into the potential impact of increased vaccination resources on the overall healthcare outcomes during the pandemic.

In conclusion, the third scenario demonstrates the importance of strategic vaccine allocation, particularly when resources are limited. Although a substantial increase in vaccine availability can lead to significant reductions in both case numbers and deaths, it is essential to prioritize the most vulnerable populations and target age groups that play a significant role in virus transmission. By understanding the potential outcomes under different prioritizations and levels of vaccine availability, policymakers can make informed decisions to optimize healthcare outcomes and mitigate the impact of the COVID-19 pandemic.

### 4.3. Discussion of Results in Context of Existing Literature

In comparing our results with those of other studies in the field, our dynamic vaccine allocation model offers a unique perspective by considering the entire United States across multiple age groups and adjusting vaccine distribution in real-time based on evolving transmission patterns. While various studies have explored optimal vaccine distribution strategies, our approach stands out in its broad scope and adaptability.

Meehan et al. (2020) [25] extended earlier models to the context of COVID-19, suggesting that prioritizing individuals aged 30–59 could effectively minimize transmission, while focusing on those aged 60 and above would reduce morbidity and mortality. Our results are consistent with these conclusions, particularly in scenarios where death reduction is prioritized. In such cases, our model also allocates more vaccines to the older population (65+ years) and, interestingly, to the younger age group (0–17 years). This aligns with the idea that protecting older adults is crucial for reducing fatalities but also highlights the potential of vaccinating younger populations to curtail transmission, which indirectly safeguards more vulnerable groups.

Similarly, Bubar et al. (2021) [18] proposed a model-informed strategy for COVID-19 vaccine prioritization, emphasizing the balance between minimizing cumulative incidence and mortality. They found that prioritizing younger adults (aged 20–49) could effectively reduce incidence, while targeting older adults could lower mortality. Our study expands on this by incorporating regional disparities and considering all age groups, thereby offering a broader perspective. Our findings suggest that optimal vaccine distribution strategies can vary significantly depending on the specific public health objectives—whether the goal is to reduce overall case numbers or to minimize deaths—and the demographic characteristics of the population.

Further, Gonzales-Parra et al. (2022) [26] explored optimal vaccination strategies using a nonlinear mathematical model, focusing on case fatality rates and population age structure. Their findings, which suggest prioritizing older adults generally leads to the best outcomes, align with our death-reduction prioritization results. However, our model’s inclusion of dynamic transmission patterns across different states adds another layer of complexity, underscoring how regional variations can significantly influence optimal vaccine distribution strategies. This aspect is particularly relevant in a large and diverse country like the United States, where transmission dynamics and healthcare infrastructure can vary widely from one region to another.

A key distinction between our study and others lies in the scope and granularity of our analysis. While many studies focus on specific age groups or regions, our model encompasses all 50 states and considers three distinct age groups. This approach allows us to account for the diverse transmission dynamics and population structures across the United States, offering a more adaptable framework for vaccine allocation. Our findings demonstrate the importance of a flexible and context-specific approach to vaccine distribution, particularly in the face of rapidly changing pandemic conditions.

In conclusion, our work contributes to the ongoing discussion on vaccine allocation strategies by offering a model and solution method that prioritize public health outcomes based on both age and region while dynamically adapting to changing circumstances. The differences observed between our results and those of other studies highlight the critical need for tailored vaccine distribution strategies that can address the unique challenges presented by regional and demographic variations during a pandemic.

## 5. Discussion and Conclusions

In this paper, we have presented an age-structured dynamic model that incorporates vaccination data, providing a more accurate and nuanced understanding of disease transmission dynamics among different age groups among multiple regions. We have also proposed a novel method for a simultaneous allocation of vaccines across regions, age groups, and time periods, taking into consideration the varying transmission severity, population structures, and limited resources for vaccine distribution. By analyzing different vaccine allocation policies under various scenarios of vaccine availability, we have provided insights into effective strategies for managing disease and mitigating impact on public health. We have also developed a tool that could be used in practice for allocating vaccines by geographic and demographic group for maximum public health benefit. By periodically running the model, allocations could be adjusted as new data become available as to rates of cases and deaths by location, age, or other demographic factors.

Our work has demonstrated the importance of data-driven and adaptable vaccine allocation policies in responding to pandemics and safeguarding public health. By incorporating regional transmission patterns, population structures, and the impact of vaccination on disease transmission and death rates, our age-structured dynamic model with vaccination and optimization framework offers a valuable tool for public health authorities and policymakers to make informed decisions on vaccine distribution.

The analysis of different vaccine allocation policies under various scenarios of vaccine availability highlights the need for targeted and strategic vaccine distribution, particularly when resources are limited. By prioritizing the most vulnerable populations and targeting age groups that play a significant role in virus transmission, it is possible to achieve substantial reductions in both cases and fatalities.

We acknowledge that while our model includes an effectiveness parameter to represent vaccine efficacy, it does not explicitly account for waning immunity over time. Given the relatively short study period, we assumed that the vaccine remained effective after administration. However, we agree that future studies could benefit from a more detailed modeling approach that explicitly represents the waning of vaccine effectiveness over time. This could involve creating separate compartments for individuals vaccinated at different times.

We recognize that no single vaccine allocation policy will be effective in all circumstances. Therefore, our emphasis is on developing a methodology for allocation. Our findings for vaccine allocation reflect the circumstances experienced in the United States for COVID-19 in early 2021. These circumstances include the exposure of individuals to the virus (a consequence of human behavior), its transmissibility, the number of vaccine doses available and administered, vaccine effectiveness and the propensity of individuals to experience severe disease. All of these factors can be time-varying. Other studies may come to different conclusions as to how vaccines might best be allocated, depending on the specific circumstances.

New human-transmissible diseases are likely to emerge in future years. When that happens, efforts will resume to create vaccines that are effective at controlling disease transmission and reducing instances of severe disease. Our age-structured dynamic model with vaccination and optimization can serve as a foundation for future research and decision making, as new data become available and new challenges arise. The ongoing refinement and application of our model will help inform evidence-based decision making for vaccine allocation and distribution, contributing to a better understanding of disease transmission dynamics and providing a more efficient and equitable use of limited resources.

In summary, our work contributes to understanding of the complex interplay between age-specific transmission dynamics, vaccine rollout, variations among regions, and disease outcomes. By developing and applying our age-structured dynamic model with vaccination, we hope to inform vaccine allocation and distribution strategies, helping to minimize the effects of infectious diseases on populations with an efficient use of limited resources.

## Figures and Tables

**Figure 1 vaccines-12-01034-f001:**
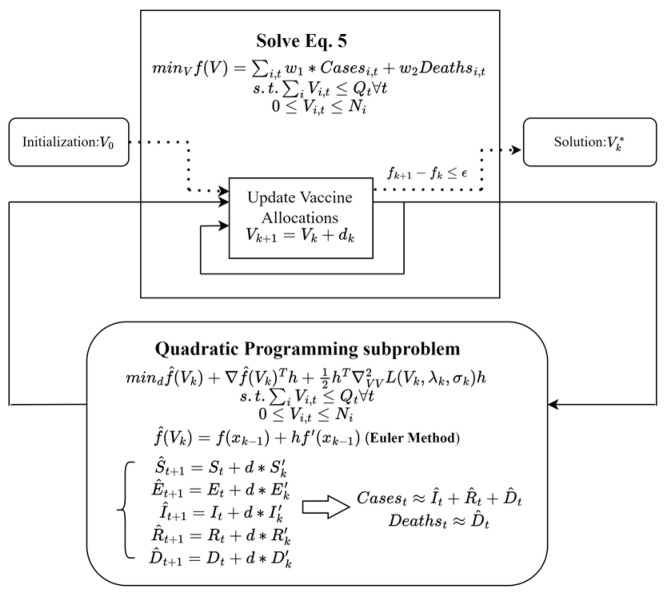
A schematic of the SLSQP algorithm.

**Figure 2 vaccines-12-01034-f002:**
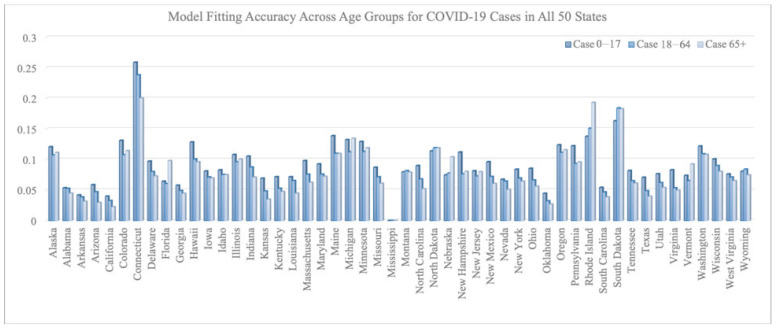
RRMSE across age groups for COVID-19 cases in all 50 states.

**Figure 3 vaccines-12-01034-f003:**
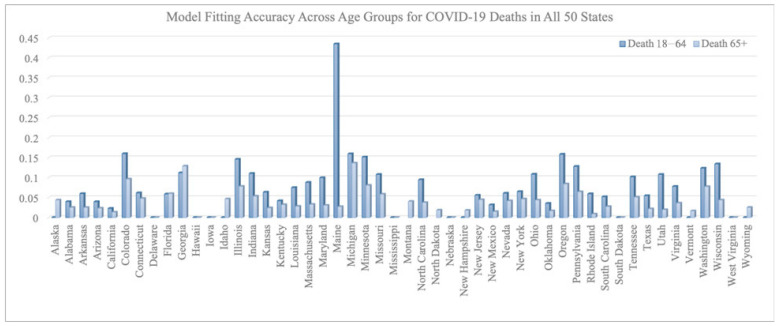
RRMSE across age groups for COVID-19 deaths in all 50 states.

**Figure 4 vaccines-12-01034-f004:**
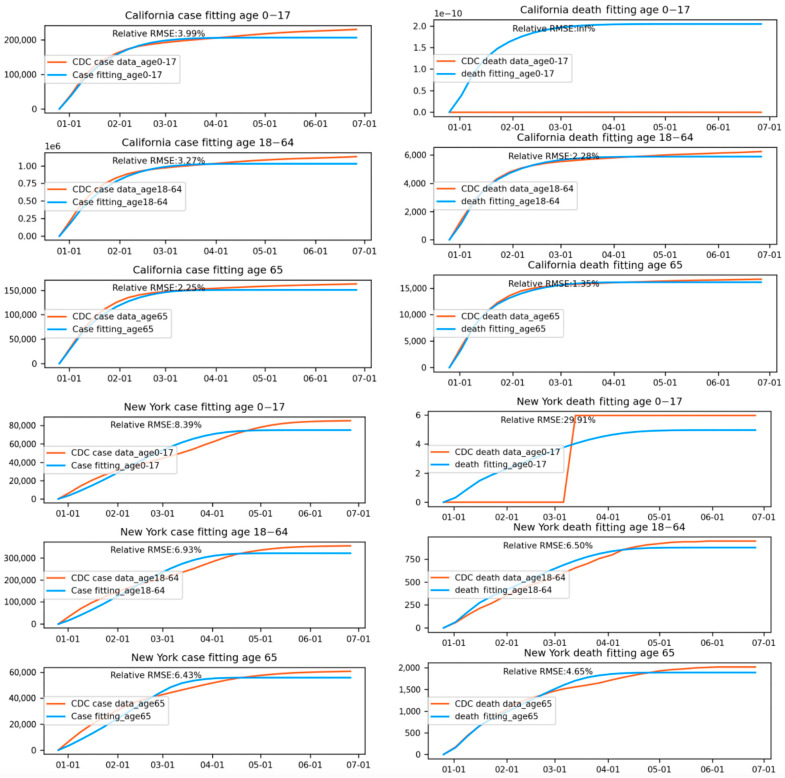
Fitting results of case/death numbers across three age groups for California and New York.

**Figure 5 vaccines-12-01034-f005:**
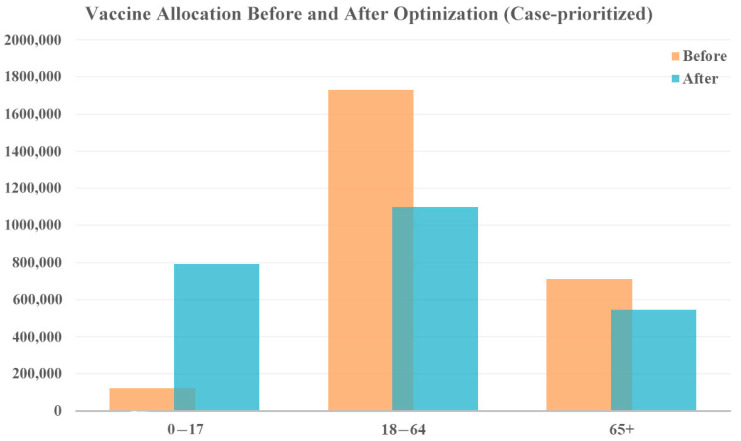
Vaccine allocation comparison for case-prioritized vaccine optimization with original vaccine availability.

**Figure 6 vaccines-12-01034-f006:**
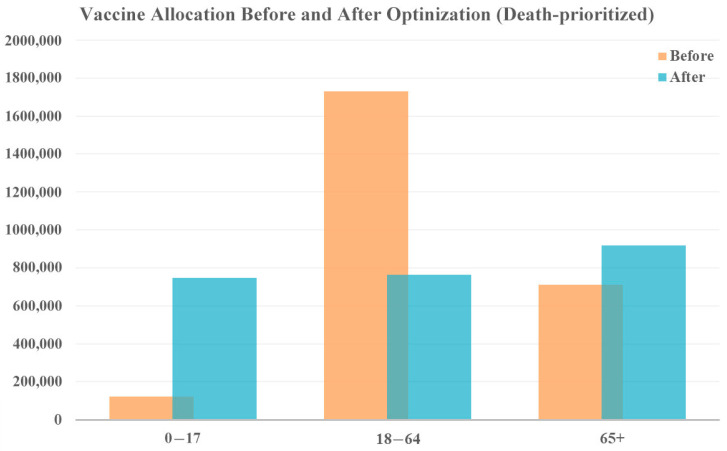
Vaccine allocation comparison for death-prioritized vaccine optimization with original vaccine availability.

**Figure 7 vaccines-12-01034-f007:**
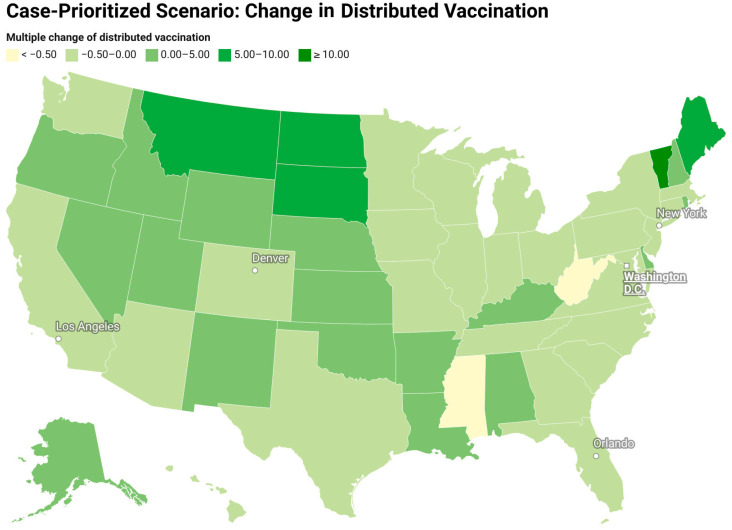
Redistribution of the original amount of vaccine among 50 states for a case-prioritized scenario (change in the vaccine distribution divided by the original amount of vaccine).

**Figure 8 vaccines-12-01034-f008:**
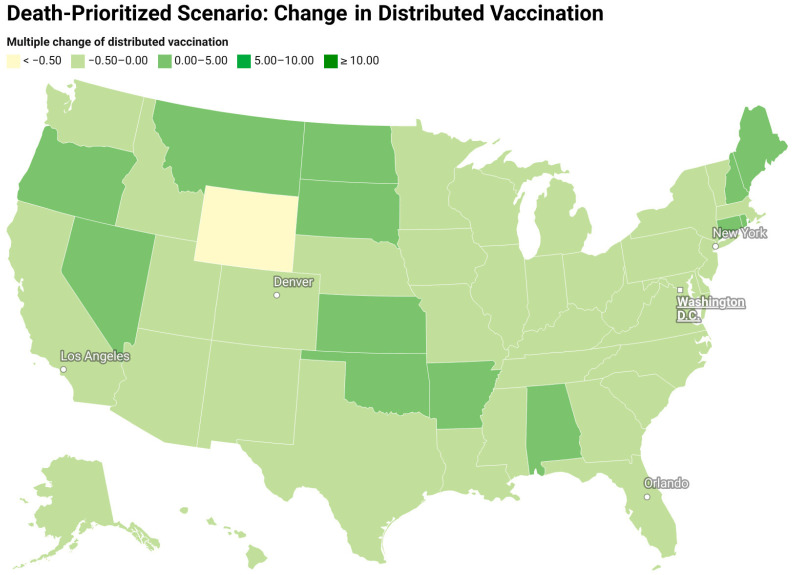
Redistribution of the original amount of vaccine among 50 states for a death-prioritized scenario (change in the vaccine distribution divided by the original amount of vaccine).

**Figure 9 vaccines-12-01034-f009:**
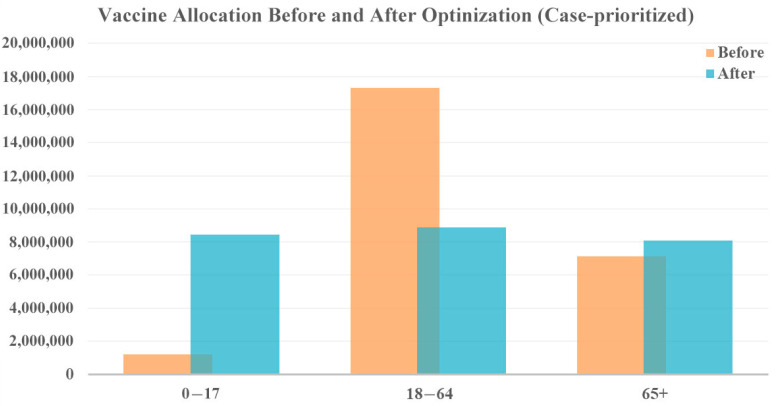
Vaccine allocation comparison for case-prioritized vaccine optimization with 10 times the vaccine availability.

**Figure 10 vaccines-12-01034-f010:**
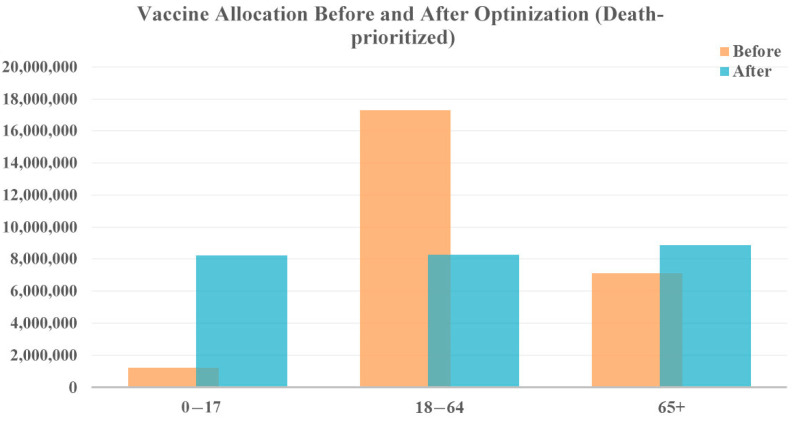
Vaccine allocation comparison for death-prioritized vaccine optimization with 10 times the vaccine availability.

## Data Availability

The datasets generated and/or analyzed during the current study are not publicly available but are available from the corresponding author on reasonable request.

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
