# Peer review of "Dynamic Vaccine Allocation for Control of Human-Transmissible Disease"

_vaccines, 2024, doi:10.3390/vaccines12091034_

Round 1
Reviewer 1 Report
Comments and Suggestions for Authors
This paper presents a methodology for vaccine allocation during pandemics, using a time-sensitive, age-structured compartmental model. Focusing on the COVID-19 pandemic in the United States, the study aims to optimize vaccine distribution across regions, age groups, and timeframes to minimize deaths and cases. The paper is well-written and relevant, but I noticed a lack of detailed mathematical formulation for the proposed control method.
I have a few points and questions for the authors:
1) There is extensive research on the decision of individuals for vaccination (usually related to game theory), which the paper did not consider. There should be a paragraph on this topic in the introduction, along with an explanation of why the authors chose not to include this factor in their model.
2) Section 3.4.2 could be improved by providing a more detailed mathematical formulation of the methodology used.
3) I also noticed the absence of a comparison between the presented results and other works in the field.
Comments on the Quality of English LanguageMinor editing of English language required.
Author Response
Thank you for the helpful feedback on our paper. We describe our edits to the manuscript in the attached file.

Reviewer 2 Report
Comments and Suggestions for Authors
This is an interesting and well written (I detected one spelling error on line 48 where very should be vary) ms. The idea of trying to mathematically optimize the allocation of vaccines is compelling. Unfortunately, a number of assumptions in their model are highly unrealistic. I name but a few.
1. The number of confirmed cases and deaths are gross undercounts.
2. mRNA vaccines protect only for a limited period against infection and death (when infected) with the period of protection against infection being much shorter than the period of protection against death. Note that this makes a realistic model essentially non-Markovian (perhaps separate compartments of those vaccinated during each moth could do the trick?) so that their modelling strategy of compartmental modeling cannot (easily) capture reality adequately.
I think that all these limitations should be acknowledged.
Author Response

(The authors gave the same response as above.)

Round 2
Reviewer 1 Report
Comments and Suggestions for Authors
The authors responded to my review, indicating that my suggestions were applied to the manuscript. While points 2 and 3 were indeed addressed, point 1 was not considered, contrary to what was stated in their reply. The authors must update the manuscript accordingly or provide a proper explanation for their decision regarding point 1.
A few papers for the authors to start:
Bauch, Chris T., and David J.D. Earn. 2004. "Vaccination and the Theory of Games." Proceedings of the National Academy of Sciences of the United States of America 101, no. 36: 13391-13394. https://doi.org/10.1073/pnas.0403823101.
Schimit, P.H.T., and L.H.A. Monteiro. 2011. "A Vaccination Game Based on Public Health Actions and Personal Decisions." Ecological Modelling 222, no. 9: n.p. https://doi.org/10.1016/j.ecolmodel.2011.02.019.
Bauch, Chris T. 2005. "Imitation Dynamics Predict Vaccinating Behaviour." Proceedings of the Royal Society B: Biological Sciences 272, no. 1573: 1669-1675. https://doi.org/10.1098/rspb.2005.3153.
Klepac, Petra, Itamar Megiddo, Bryan T. Grenfell, and Ramanan Laxminarayan. 2016. "Self-Enforcing Regional Vaccination Agreements." Journal of the Royal Society Interface 13, no. 114. https://doi.org/10.1098/rsif.2015.0907.
Reluga, Timothy C., Chris T. Bauch, and Alison P. Galvani. 2006. "Evolving Public Perceptions and Stability in Vaccine Uptake." Mathematical Biosciences 204, no. 2: 185-198. https://doi.org/10.1016/j.mbs.2006.08.015.
Arefin, Md Rajib, K. M.Ariful Kabir, Marko Jusup, Hiromu Ito, and Jun Tanimoto. 2020. "Social Efficiency Deficit Deciphers Social Dilemmas." Scientific Reports 10, no. 1. https://doi.org/10.1038/s41598-020-72971-y.
Fu, Feng, Daniel I. Rosenbloom, Long Wang, and Martin A. Nowak. 2011. "Imitation Dynamics of Vaccination Behaviour on Social Networks." Proceedings of the Royal Society B: Biological Sciences 278, no. 1702: 42-49. https://doi.org/10.1098/rspb.2010.1107.
Comments on the Quality of English LanguageMinor editing of English language required.
Author Response
Dear Reviewer,
We appreciate your insightful comments and suggestions, which have helped enhance the clarity and depth of our manuscript. Below, we address each of your comments specifically.
Comment 1: The authors responded to my review, indicating that my suggestions were applied to the manuscript. While points 2 and 3 were indeed addressed, point 1 was not considered, contrary to what was stated in their reply. The authors must update the manuscript accordingly or provide a proper explanation for their decision regarding point 1.
Response:
We appreciate the reviewer's emphasis on the significance of individual decision-making in vaccination behavior. Initially, our study was conducted during a period when the most pressing challenge was the shortage of vaccines, rather than a lack of willingness among individuals to get vaccinated. Therefore, our model focused on optimizing the allocation of limited vaccine resources across different regions and age groups, assuming that the available vaccines would be fully utilized.
However, we acknowledge the importance of individual decision-making in broader vaccination strategies. To address this, we have revised the introduction to include a discussion on how individual vaccination decisions can impact public health outcomes in lines 53-78. We have also explained why these factors were not included in our model, given the specific context and objectives of our study. We understand that individual decision-making is a critical factor in other contexts, and future work could extend our model to incorporate these dynamics.
Thank you for your valuable feedback. We hope we have addressed all your comments thoroughly and appreciate your guidance throughout the revision process.

Round 3
Reviewer 1 Report
Comments and Suggestions for Authors
The authors have addressed all the feedback provided during the review process, enhancing the article.